# HMGA1 Is a Potential Driver of Preeclampsia Pathogenesis by Interference with Extravillous Trophoblasts Invasion

**DOI:** 10.3390/biom11060822

**Published:** 2021-05-31

**Authors:** Keiichi Matsubara, Yuko Matsubara, Yuka Uchikura, Katsuko Takagi, Akiko Yano, Takashi Sugiyama

**Affiliations:** 1Department of Regional Pediatrics and Perinatology, Graduate School of Medicine, Ehime University, Ehime, Toon-shi 791-0295, Shitsukawa, Japan; 2Department of Obstetrics and Gynecology, School of Medicine, Ehime University, Ehime, Toon-shi 791-0295, Shitsukawa, Japan; takeyu@m.ehime-u.ac.jp (Y.M.); yuka.itani@gmail.com (Y.U.); takagi-k@m.ehime-u.ac.jp (K.T.); a.23-yano@hotmail.co.jp (A.Y.); sugiyama@m.ehime-u.ac.jp (T.S.)

**Keywords:** preeclampsia, HMGA, immunotolerance, extravillous trophoblast, placentation

## Abstract

Preeclampsia (PE) is a serious disease that can be fatal for the mother and fetus. The two-stage theory has been proposed as its cause, with the first stage comprising poor placentation associated with the failure of fertilized egg implantation. Successful implantation and placentation require maternal immunotolerance of the fertilized egg as a semi-allograft and appropriate extravillous trophoblast (EVT) invasion of the decidua and myometrium. The disturbance of EVT invasion during implantation in PE results in impaired spiral artery remodeling. PE is thought to be caused by hypoxia during remodeling failure–derived poor placentation, which results in chronic inflammation. High-mobility group protein A (HMGA) is involved in the growth and invasion of cancer cells and likely in the growth and invasion of trophoblasts. Its mechanism of action is associated with immunotolerance. Thus, HMGA is thought to play a pivotal role in successful pregnancy, and its dysfunction may be related to the pathogenesis of PE. The evaluation of HMGA function and its changes in PE might confirm that it is a reliable biomarker of PE and provide prospects for PE treatment through the induction of EVT proliferation and invasion during the implantation.

## 1. Introduction

The high-mobility group (HMG) of chromosomal proteins regulates DNA-dependent biochemical processes [1]. HMG proteins are important components of the enhanceosome, which is an enhancer that regulates gene expression [2]. The enhanceosome is involved in protein–protein interactions occurring within it, and in the regulation of chromatin, resulting in DNA transcription, replication, and recombination. It is also involved in many intracellular functions (e.g., cellular proliferation, invasion, angiogenesis, and immune tolerance) and reduces apoptosis [3]. HMG proteins are classified into three superfamilies (HMGA, HMGB, and HMGN), each with a characteristic functional domain (the AT hook, HMG box, and nucleosome-binding domain, respectively). Proteins containing any of these three functional sequences are referred to as HMG motif proteins.

HMG proteins are involved in autoimmune diseases [4,5]. HMGB1 has anti-DNA effects on rheumatoid arthritis, systemic lupus erythematosus (SLE), and myositis through the formation of immunostimulatory complexes with proinflammatory cytokines. It is thought to be an important autoimmune disease mediator and biomarker. Tzioufas et al. [5] reported the detection of HMG-17 antibodies in serum from patients with many different autoimmune diseases, including SLE, and concluded that HMG-17 is associated with SLE activity.

HMGA has HMGA1 and HMGA2 forms. HMGA1 is a 20-kDa protein with three AT hooks, which are DNA-binding domains. It has been reported to activate transcription of the insulin receptor [6]. It binds to the insulin response element of the insulin-like growth factor-I (IGF-I)/IGF binding protein 1 (IGFBP1) gene promoter, interfering with the inhibitory effect of insulin on IGFBP1 gene transcription [7]. HMGA1 is involved in cellular activation, including that related to insulin activity and cancer cell proliferation. It is expressed particularly strongly in cancer cells [8] and is very likely to be involved in the pathogenesis of preeclampsia (PE). PE entails insulin resistance and the inhibition of trophoblast (physiological cancer cell) invasion at the implantation site (necessary for the maintenance of normal pregnancy), resulting in the disturbance of spiral artery remodeling.

## 2. Pathophysiology of PE

PE is a hypertensive disorder of pregnancy that is associated with proteinuria and/or organ failure and is a major cause of maternal and neonatal death. PE is characterized by vascular constriction, which results in maternal organ failure, and increased vascular resistance of the placental and systemic circulation, which leads to reduced uteroplacental blood flow followed by placental dysfunction [9] and fetal growth restriction (FGR) [10]. The impairment of circulatory homeostasis is caused mainly by vascular endothelial dysfunction [11]. The systemic blood vessels constrict easily and lose anticoagulant activity in PE [12]. The hypervascular contractility and hypercoagulability seen in patients with PE are thought to be caused by chronic placental inflammation. This inflammation may be attributable to increased placental levels of proinflammatory cytokines, including tumor necrosis factor-α, and reactive oxygen species, or to placental ischemic changes associated with poor placentation [12,13]. In addition to the overproduction of proinflammatory cytokines and oxidative stress, soluble fms-like tyrosine kinase 1 and soluble endoglin, which are anti-angiogenic factors that inhibit angiogenesis and vasorelaxation, are overproduced due to poor placentation in PE, resulting in increased vascular contractility and coagulation.

Skjærven et al. [14] reported that PE may be triggered by interaction between maternal genes and fetal genes from the father. Maternal and paternal factors contribute to the risk of PE; the relationship between trophoblasts with paternal genes and decidual components with maternal genes is crucial for its pathogenesis. According to Redman and Sargent’s [15] widely accepted two-stage theory, the etiology of PE can be explained by poor placentation caused by impaired spiral artery remodeling during implantation in early pregnancy (the first stage), which causes systemic organ failure via the ischemic placenta–derived secretion of proinflammatory cytokines and anti-angiogenic factors into the systemic circulation (the second stage; Figure 1). Early in pregnancy, cytotrophoblasts (CTs) covering the surfaces of blastocysts form cell columns that attach to the maternal decidua and give rise to an extravillous trophoblast (EVT) lineage. These EVTs invade the decidua, reaching the uterine spiral artery, and differentiate into endovascular CTs that cover the spiral artery lumina. This process results in remodeling, in which spiral artery converts into large ducts without muscle cells around the vessels to send abundant maternal blood to the placenta (Figure 1). Early in normal pregnancy, the invasion of EVTs as a semi-allograft is induced by the creation of an immunotolerant environment in cooperation with decidual natural killer (NK) cells and macrophages in the decidua. However, immune tolerance is disrupted by the activation of an immune response in the primary PE lesion, and chronic inflammation associated with increased proinflammatory cytokine levels occurs in the decidua, leading to disturbed spiral artery remodeling. Overall, the earliest and most important change that occurs during PE pathogenesis is the activation of the maternal immune response to EVTs, which is associated with the disruption of maternal immune tolerance and impaired EVT invasion of the uterine decidua and myometrium, resulting in disturbed spiral artery remodeling.

## 3. HMGA1

HMGA has three AT hooks that bind AT-rich DNA; it modulates the chromatin conformation of target DNA and has HMGA1 and HMGA2 forms synthesized by alternative splicing [16]. HMGA1 is further classified into three proteins (HMGA1a–c) that influence many biological processes, including cell growth, proliferation, differentiation, and death [17]. HMGA1 is localized in the nucleus and predominantly in condensed chromatin. The Hmga1 gene is expressed essentially in all tissues, but particularly highly in undifferentiated cells, such as embryonic cells [18], reflecting its critical roles in embryogenesis and organ development [19]. Pierantoni et al. [20] suggested that HMGA1′s regulation of SAC gene expression contributes to the maintenance of genomic stability in embryonic cells. HMGA1 deficiency and overexpression cause B-cell, T-cell, and NK-cell lymphoma in mice [21,22]. Such HMGA1 abnormality reduces the immune response by promoting the differentiation of immature T cells to regulatory T cells, which suppress other proinflammatory T cells, including T helper (Th)17 cells. The immunosuppressive effect of HMGA1 might be favorable for the invasion of EVT invasion during placentation. On the other hand, the hmga1 gene is overexpressed in several types of cancer [23]. Constitutive HMGA overexpression correlates with tumor growth and increased metastasis, and thus poor patient prognoses [24].

Fedele et al. [25] reported that HMGA1 in human thyroid cells exerts opposite effects on the growth of neoplastic and normal cells (neoplastic transformation and apoptosis, respectively). Strong HMGA expression in normal cells promotes apoptosis of these cells via activation of the caspase-3 pathway [25]. Recently, HMGA was also implicated in the senescence of normal cells such as fibroblasts [20]. HMGA1b, induces apoptosis in normal thyroid cells, but is not involved in neoplastic transformation [25]. Williams et al. [26,27] reported that HMGA1 was expressed more strongly in colorectal cancer than in nonmalignant colonic epithelium and suggested that it drives metastatic progression by inducing genes involved in epithelial–mesenchymal transition (EMT) and promoting stem cell properties.

Furthermore, the up-regulation of matrix metalloproteinase-2 (MMP-2) by HMGA1 promotes lung cancer transformation, and the blocking of MMP-2 expression inhibits cancer cell migration and invasion [28]. HMGA1 is also a central factor in tumor progression in patients with triple-negative breast cancer [29] and pancreatic cancer [30]. It has been reported to activate the expression of stem-cell transcriptional networks, which are involved in tumor progression and EMT [31,32]. High levels of HMGA1 expression lead to poor prognosis in patients with malignant tumors [24]. HMGA1 is especially involved in the proliferation and invasion of activated cells, such as cancer cells. In contrast, HMGA2 is thought to inhibit the proliferation of stable normal cells via apoptosis.

## 4. HMGA2

HMGA2 is a non-histone architectural transcription factor that alters the chromatin structure and regulates gene transcription. It is known to influence processes such as the cell cycle, DNA damage repair, apoptosis, senescence, and EMT [33]. HMGA2 overexpression is thought to be a feature of cancer, and increases in HMGA2 expression are used to predict the efficacy of some cancer chemotherapies. HMGA2 also plays a critical role in embryonic stem cell development and its dysregulation in adult somatic cells can lead to carcinogenesis; mutation of the HMGA2-encoding gene is observed widely in diverse types of tumor [34,35]. HMGA2 is expressed strongly in embryonic stem cells during embryogenesis and in malignancies. Thus, it may be required for the fetal development and carcinogenesis. A variant of HMGA2 is associated with human height and FGR via cell growth reduction [36,37]. HMGA2 regulates IGF2 expression for physiological fetal and postnatal growth. On the other hand, fibroblasts with HMGA2 overexpression have been reported to cause fibrosarcoma into athymic nude mice with distant metastases [38]. Since HMGA2 positively correlates with distant metastasis and poor survival rates HMGA2 could be available as a potential diagnostic and prognostic tumor marker [39].

Transforming growth factor-β (TGF-β) is important for appropriate placentation, and its dysfunction may be related to the pathogenesis of PE [40]. It can induce the transcription of HMGA2, and TGF-β–induced Smad4 binds directly to the HMGA2 promoter to regulate the EMT caused by TGF-β during embryogenesis and related to carcinoma and EVT invasion [41]. Recently, Yu et al. [42] reported that increases in HMGA2 expression enhanced the proliferation and reduced the aging of human umbilical cord blood–derived stromal cells, and was accompanied by increased cyclin E and cell division cycle 25A expression and decreased cyclin-dependent kinase inhibitor expression. HMGA2 is known to be involved in cell cycle regulation through cyclin A2, a transcriptional regulator [43]. It plays an important role in chromosome condensation during the meiotic G2/M transition and myogenesis, resulting in satellite cell activation [44]. Satellite cells are progenitor cells with little or no cytoplasm found in mature muscle [45]; when activated, they proliferate and differentiate into myoblasts [46]. Hmga2 is expressed in satellite cells most strongly during early embryogenesis and is rarely expressed in late pregnancy. During myoblast proliferation and early myogenesis, HMGA2 is up-regulated along with satellite cell activation [47], and knockout of the *hmga2* gene results in skeletal muscle damage [35,47].

## 5. Roles of HMGA1 and HMGA2 in Pregnancy

HMGA2 plays an important role in embryogenesis; it is highly expressed in murine and human fetal tissues, but not in adult tissues, with the exception of the uterine myometrium [35,48]. HMGA2 is involved in mammalian growth and development, and its absence results in fetal growth retardation [35]. HMGA1 gene expression is maximal during embryonic development, then decreases and becomes restricted to certain tissues over a period of days [19]. Recently, Bamberger et al. [49] demonstrated that HMGA1 was located specifically in the nuclei of villous CTs and intermediate EVTs of the human placenta with highly proliferative activity, but not in villous syncytiotrophoblasts (STs) with low proliferative activity. Furthermore, its expression was especially strong in anchoring villi at the implantation site and in EVTs inside the maternal decidua. Together with MMP-9, HMGA1 affects cellular motility [50], and it may be involved strongly in EVT invasion, which is important for normal placentation. Briese et al. [51] also reported that HMGA1 was highly expressed in villous CTs, but not villous STs, in complete hydatidiform moles, and demonstrated that it was highly expressed at the invasion sites of invasive moles and choriocarcinoma. Thus, HMGA1 and HMGA2 play important roles in trophoblast proliferation and invasion and are thought to be critical for placentation and fetal development.

West et al. [52] reported on the mechanism of HMGA2′s involvement in trophoblast proliferation and differentiation. In early pregnancy, placental micro-RNA (miR)-182 levels are high to prevent BRCA1 translation, leading to high HMGA2 levels. Highly expressed HMGA2 can be involved in the differentiation of CTs into STs. Thereafter, miR-182 levels decrease, stimulating the formation of the BRCA1/CtIP/ZNF350 repressor complex, which binds to HMGA2, preventing transcription [52]. miRNAs regulate the functions of various cells and are involved in the pathogenesis of many diseases. Placenta-derived miRNAs can be detected in maternal plasma [53], and these miRNAs have been used for the diagnosis of various diseases and prenatal conditions. Pregnancy-related miRNA clusters (C19MC, C14MC, and miR-371-3) are involved in physiological changes in normal pregnancy and in the pathogenesis of various pregnancy complications [53]. West et al. [52] also reported that let-7 miRNAs are inhibited by the RNA-binding proteins LIN28A and LIN28B. This inhibition plays an important role in early embryogenesis by stabilizing HMGA2 expression. HMGA2 is regulated by the BRCA1/ZNF350/CtIP repressor complex, and miRNA-182 can prevent BRCA1 translation and increase HMGA2 expression in cancer cells. Interaction between BRCA1 and HMGA2 may be necessary for early human placental development. Ali et al. [54] reported that high LIN28A and LIN28B expression levels, which reduce let-7 miRNA expression, stimulated the expression of HMGA1 in trophoblasts through increased ARID3A, ARID3B, and KDM4C complex expression. The knockdown of LIN28A- or LIN28B-derived high let-7 miRNA expression impairs placentation, resulting in FGR and spontaneous abortion [55]. FGR may also be caused by reduced CT production due to the placental expression of LIN28A and LIN28B or ARID3A and ARID3B. Such HMGA2related mechanisms have also been associated with mammalian height [36] and organ size [56,57,58].

On the other hand, Bamberger et al. [49] found that HMGA1 was expressed in the nuclei of highly proliferative CTs, but not in the nuclei of terminally differentiated STs, in the placenta; it is strongly expressed in EVTs with strong migration capacity. They concluded that the nuclear expression of HMGA1 was important for the function of EVTs, which infiltrate the maternal decidua and myometrium, and participate in spiral artery remodeling and appropriate placentation [49]. EVT invasion requires microenvironmental preparation via the degradation of stromal collagen by proteases such as MMPs. HMGA1 can promote cell growth and invasion by activating MMP-2 during placentation [59]. The activation of MMP-2 and MMP-9 by HMGA1 and the interaction of these MMPs are reportedly important for the growth and invasion of cancer cells in malignant tumors [60,61] and those of EVTs in the placenta [62]. Furthermore, HMGA2 messenger RNA expression in placental tissue is markedly elevated in early pregnancy [63]. Thus, HMGA1 and HMGA2 are thought to play important roles in placenta and embryo development in early pregnancy.

Xian et al. reported that HMGA1 could amplify Wnt/β-catenin signaling [64]. It is known that activation of Wnt/β-catenin is required for HMGA1 to promote tumor cell invasion [65], and conversely, Wnt/β-catenin is involved in tumor cell metastasis by promoting HMGA1 expression [66]. On the other hand, it has been reported that microRNA-758 simultaneously suppresses HMGA1 and Wnt/β-catenin when it inhibits tumor cell growth [67], suggesting that the HMGA1/Wnt/β-catenin system may also be involved in the growth and invasion of EVTs. Furthermore, HMGA1 could promote cell differentiation by stimulating SOX9 [64], which may be involved in the differentiation of EVTs into endovascular trophoblasts during placentation.

## 6. The Role of HMGA in the Pathogenesis of PE

Although HMGAs are expressed strongly in the placenta and are thought to be involved in placentation and the regulation of trophoblast function, the expressions of HMGA1 and HMGA2 in the placenta differ markedly [63]. HMGA1 is expressed exclusively in trophoblasts, whereas HMGA2 is expressed strongly in the stromal cells of placental villi. Both are expressed strongly in nuclei and weakly in cytoplasm. The functions of these two HMGAs in the placenta are thought to complement each other during placentation for the creation of an optimal microenvironment for pregnancy. Thus, their dysfunction can lead to PE pathogenesis (Figure 2).

Interferon (IFN)-β is an immune modulator that promotes tolerance of paternal antigens at the maternal–fetal interface [68], and HMGA1 is involved in its function [69]. As the pathogenesis of PE involves disturbance of the immunotolerance of embryos (which contain paternal antigens), HMGA1 may be related to it through IFN-β dysfunction. IFN-γ also influences the disturbance of immunotolerance during pregnancy. It is produced by Th1 and is associated with the pathogeneses of spontaneous abortion and PE [70]. Liu et al. [71] reported that IFN-γ inhibited EVT migration and induced apoptosis by increasing phosphorylated JAK/STAT1 and caspase 3 expression and decreasing platelet-derived growth factor receptor A expression. HMGA1 regulates IFN-γ gene expression [72], as it is involved in Th cell differentiation via the regulation of cytokines such as IFN-γ [73]. HMGA1 is likely to participate in Th1-specific gene expression and promote enhanceosome formation on the Th1 cytokine gene. Hopper et al. [74] reported that increased HMGA1 expression in pulmonary arterial endothelial cells as a result of dysfunctional bone morphogenetic protein receptor-2 signaling transited endothelium to smooth muscle–like cells, leading to pulmonary arterial hypertension. HMGA1 is thought to regulate immune and inflammatory reactions and be involved in endothelial dysfunction–based hypertension.

In a PE mouse model used to investigate the mechanism of HMGA1′s involvement in the pathogenesis of PE, we demonstrated that HMGA1 played an important role in the regulation of EVT migration and proliferation [75]. In normal pregnancy, HMGA1 is expressed only in the nuclei of trophoblasts; we observed its extracellular release and cytoplasmic expression in trophoblast giant cells, especially in early pregnancy, in our PE model (Figure 3) [75]. We also observed HMGA1 expression around the fertilized egg during implantation. Such HMGA1 expression in cytoplasm and extracellularly was also observed in human placentas with PE. In our study, an MTT assay demonstrated that HMGA1 overexpression stimulated EVT proliferation compared with a negative control, and a transwell migration assay showed that HMGA1 stimulated EVT migration significantly compared with a control. The translocation of nuclear HMGA1 to cytoplasm treated with deoxycholic acid inhibited the cell migration significantly, suggesting that proper subcellular localization of HMGA1 is important for its functioning of HMGA1 during the EVTs migration into the decidua and myometrium in normal pregnancy. Our results suggest that HMGA1 is necessary for the maintenance of normal pregnancy via the promotion of cell proliferation and nuclear EVT invasion in the nuclei of EVTs, and that such physiological effects of EVTs are impaired when HMGA1 is secreted into the cytoplasm or extracellular space, resulting in the development of PE caused by poor placentation through impaired spiral artery remodeling.

In addition to HMGA, there are other HMG-proteins that reported to be associated with the pathogenesis of PE. One of them is HMGN1, which encodes a protein associated with active chromatin and transcription. HMGN1 could stimulate cytokine production and be capable of stimulating immune tolerance by a TLR4-dependent pathway [76]. Furthermore, HMGN1 is reportedly increased in the decidua early in pregnancy and HMGN1 is thought to be involved in the decidualization through the differentiation of uterine stromal cells [77]. On the other hand, Ducat et al. demonstrated that HMGN1 was significantly down-regulated in preeclamptic placenta [78]. In addition, STOX1A and STOX1B stimulated the expression of HMGN1 in JEG3 overexpressing STOX isoforms; however, HMGN1 was reduced in BeWo cells [79]. HMGN1 might to be involved in the physiology of normal pregnancy and the pathophysiology of PE through modulating immune tolerance.

HMGB1 is a cytokine mediator of inflammation and known as to be released from macrophages and dendric cells [80]. In particular, HMGB1 was reported as a potential blood marker for PE because it is increased in the blood reflecting the involvement of chronic inflammation in the pathogenesis of PE [81].

## 7. Conclusion

HMGAs are thought to play an important role in normal placentation by creating an optimal microenvironment for spiral artery remodeling and EVTs invasion; disturbance of these processes may play a major role in the pathogenesis of PE through poor placentation resulting in the reduction of uteroplacental perfusion. In previous work, we found that HMGA1 translocation from trophoblast nuclei to the cytoplasm contributed to the impairment of EVT invasion [75]. Thus, HMGA1 is a potential driver of PE pathogenesis via interference with EVT invasion.

## Figures and Tables

**Figure 1 biomolecules-11-00822-f001:**
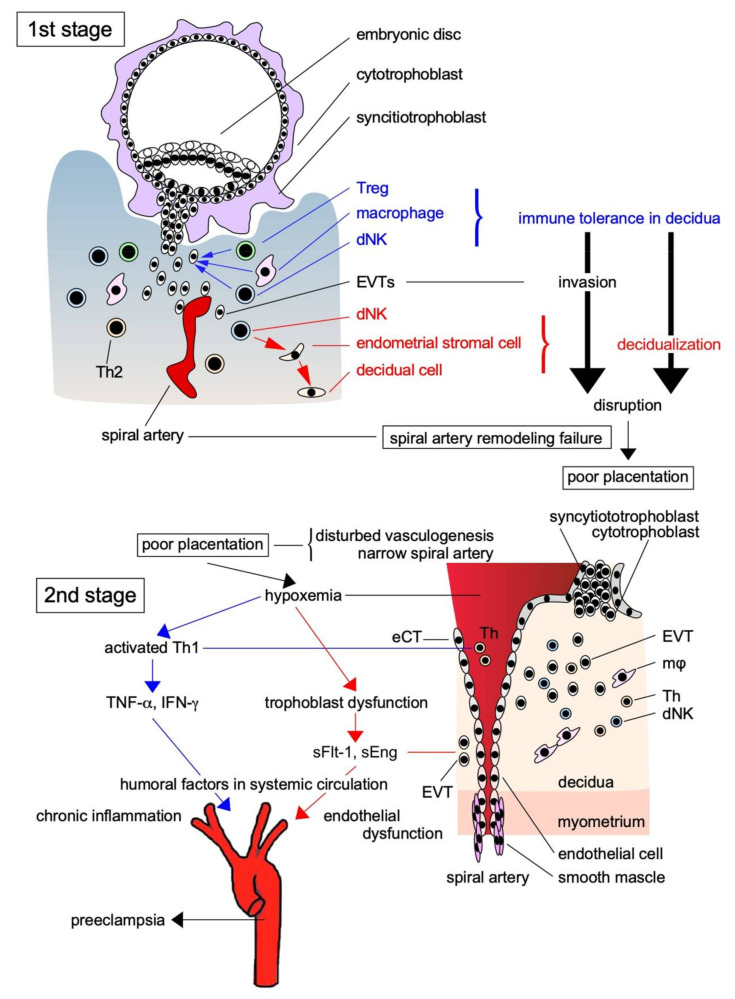
Two-stage theory of the pathogenesis of PE. The pathogenesis of PE is explained by the two-stage theory: in the first stage, EVTs as the paternally antigenic fetal components infiltrate toward the spiral artery without being eliminated by the maternal immune system due to immune tolerance by Treg and other immune cells, and supply a large amount of maternally derived blood to the placenta resulted in the successful placentation. However, when placentation is impaired by the breakdown of immune tolerance, anti-angiogenic factors and proinflammatory factors produced by placental ischemia in the second stage are released into the systemic circulation, causing multiple organ failure due to the vascular injury. Treg: regulatory T cell; dNK: decidual natural killer cell; Th: helper T cell; sFlt-1: soluble fms-like tyrosine kinase-1; sEng: soluble endoglin; TNF-α: tumor necrosis factor-α; INF-γ: interferon-γ; eCT: endovascular cytotrophoblast.

**Figure 2 biomolecules-11-00822-f002:**
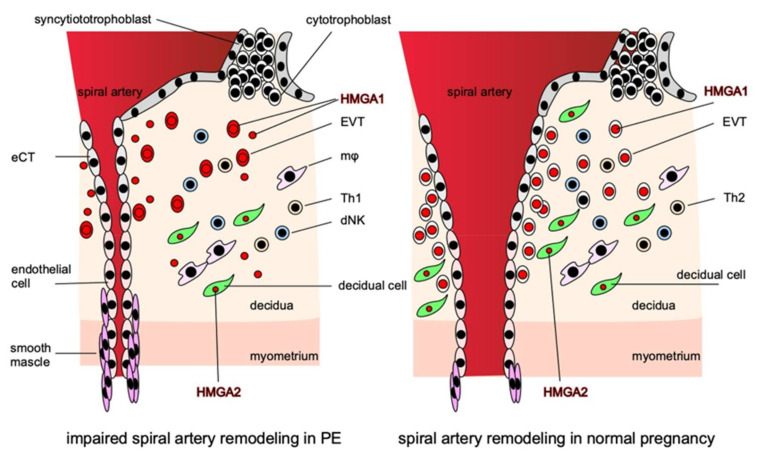
HMGAs are involved in the pathogenesis of PE. HMGA1 is localized in the nuclei of EVTs and is involved in their proliferation and invasion, while in PE, HMGA1 is released from the nuclei of EVTs into the cytoplasm and extracellular space from early pregnancy and impairs EVTs proliferation and invasion. HMGA2 is localized in the nuclei of decidual cells and, together with HMGA1, creates an appropriate microenvironment that promotes the proper invasion of EVTs at the site of implantation. In the disturbance of proper EVTs invasion, the remodeling of the spiral arteries is disturbed, resulting in narrow vessels, and the perivascular smooth muscle cannot supply sufficient blood to the placenta by constricting the vessels, thus inhibiting placentation.

**Figure 3 biomolecules-11-00822-f003:**
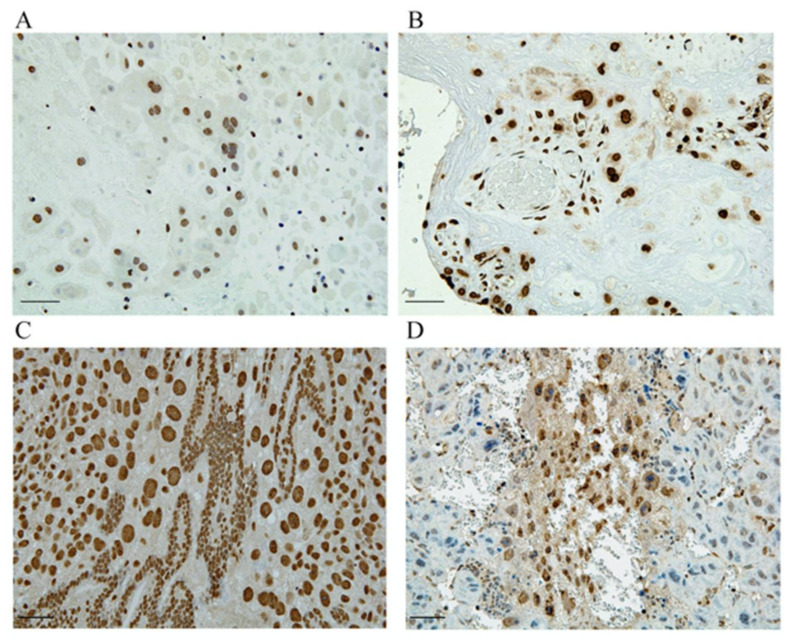
HMGA1 is expressed in the nuclei of trophoblasts. (**A**,**C**) HMGA1 is expressed only in the nuclei of EVTs derived from human normal placenta and at the implantation site of normal pregnant mouse. (**B**,**D**) HMGA1 in cytoplasm is observed in human PE placenta and HMGA1 is also extracellularly released from trophoblasts derived from our PE model. Scal bar = 100um. (This figure is quoted and modified from Ref [75]).

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
