# Peer review of "HMGA1 Is a Potential Driver of Preeclampsia Pathogenesis by Interference with Extravillous Trophoblasts Invasion"

_biomolecules, 2021, doi:10.3390/biom11060822_

Round 1

Reviewer 1 Report

In this paper by Matsubara and coworkers, the authors review the involvement of HMGA in the pathophysiology of preeclampsia. The starting point of this review seem to be a 2017 paper by the authors linking HMGA proteins with extra-villous trophoblast function.

The authors introduce HMG proteins, then describe some aspects of the features of preeclampsia, under the widely accepted theory of a two-step disease. Then, HMGA1 is further described, with its regulation and some of its known targets. In this description, some additional information would have been useful, such as the size of the protein, which is a specific feature by itself, the expression level in various cell types, as well in specific tissues like the placenta. The same remarks applied to HMGA2, described in paragraph 4, and the question is interesting since HMGA2 seems to be much less expressed in the placenta according to ProteinAtlas database. Then the role of HMGA1 and HMGA2 in pregnancy is described, and the role of HMGA proteins in preeclampsia.  

In the figure 2, the authors present the expression of HMGA1/A2 in the decidua, hypothesizing an important role of artery remodeling. In fact, the results of the authors tend to demonstrate rather a role in extravillous trophoblast invasion, a process which may not be associated to artery remodeling, which needs a specific change in trophoblast towards an endothelial-like phenotype. It seems that the direct involvement of HMGA in this transition is not well established, while its role in proliferation and cell division seems to be major as also reported by the authors.

One concern is that HMG are ubiquitous proteins involved in very general processes. So, the question is maybe that such review could be written for any protein having a general role in cell physiology; i.e. trophoblast proliferation is important in placental function, ergo, any gene playing a role on the cell cycle will have an effect on placental health. Are there placental-specific co-factors interacting with HMGA proteins?

One way of enhancing this review would be to extend it to other HMG factors that are well expressed in the placenta and placental cells, or somehow connected with placental pathophysiology. For instance, a recent study, pointed to HMGN1 as being targeted by the preeclampsia-associated transcription factor STOX1 (Ducat et al, iScience, 2020). The authors should also cite (Li et al, Int J Obstet and Gynecol 2018 PMID: 29569400, that shows that HMGB1 is increased in preeclampsia sera, as well as other papers, that are anyway not very numerous.

In fact, the title stating that HMGA are key molecules for pregnancy and preeclampsia is a bit far-reaching. It is true, but would probably be true for any ubiquitously acting protein. The same title could be made using ACTIN, GAPDH, TBP or others instead of HMGA, and it would still be true. In their last sentence, the authors state that ‘HMGA1 is a potential driver of PE…’, this could be a title much more acceptable.  

Author Response

Thank you for the thoughtful and constructive feedback you provided regarding our manuscript, “HMGA is a key molecule for the physiology of normal pregnancy and the pathogenesis of preeclampsia”. We agree with you that suggested to change and modulate my manuscript, and we have amended this by changing the contents in order to better conform

with the formatting and content rules of Biomolecules.

Reply on the 1st reviewer,

#1. “In their last sentence, the authors state that ‘HMGA1 is a potential driver of PE…’, this could be a title much more acceptable.”

              Ans1. Thank you for your advice. We change the title from original sentence to revised sentence “HMGA1 is a potential driver of preeclampsia pathogenesis by interference with extravillous trophoblasts invasion.”.

#2. “It seems that the direct involvement of HMGA in this transition is not well established”

              Ans2. Thank you for your suggestion. It is true and we have included new sentences concerning to the transition in the last paragraph.

#3. “Are there placental-specific co-factors interacting with HMGA proteins?”

              Ans3. Thank you for your suggestion. It is known that the intracellular signaling pathway of Wnt plays an important role in the function of HMGA1. We have added that explanation into the manuscript.

#4. “a recent study, pointed to HMGN1 as being targeted by the preeclampsia-associated transcription factor STOX1 (Ducat et al, iScience, 2020).”

Ans4. We cited the reference and added new sentences about HMGN1.

#5. “The authors should also cite (Li et al, Int J Obstet and Gynecol 2018 PMID: 29569400, that shows that HMGB1 is increased in preeclampsia sera”

              Ans5. We also cited the reference.

Reviewer 2 Report

The manuscript submitted by Matsubara and colleagues presents a thorough overview of the role of HMGA in normal pregnancy and preeclampsia. Overall the manuscript is well written, however there are a few points to address.

  1. On line 50-51, the authors refer to trophoblasts as physiological cancer cells. This may be misleading to some readers and confused for choriocarcinoma cells. I would suggest re-wording this.
  2. Throughout the text, the authors list roles for HMGA in various cellular processes, but only superficially. For example, on lines 152-153 the authors state that HMGA2 plays an important role in chromosome condensation during the meiotic G2/M transition and myogenesis, but they do not elaborate on how HMGA2 accomplishes this. It would be more informative if the authors could provide more details. Some other examples on lines 140-142, lines 202-204, lines 241-242.
  3. On line 239, the authors describe their own work, referring to their PE mouse model. Please elaborate on the details of this mouse model and the data that were generated from it.
  4. Figure 1 provides a nice overview of the two-stage theory of PE, but the images on the right half of the figure do not portray what is described on the left half. Specifically, the left side describes disruption of EVT invasion and decidualization, immune involvement, endothelial dysfunction, chronic inflammation and release of humoral factors to the maternal circulation. None of these features are depicted in the images on the right.
  5. Figure 2 also does not depict the situation described in the legend. Visually the two images are very similar, with only subtle differences that are quite difficult to notice. Please accentuate the differences between the normal and PE situation in the images. For example EVT proliferation and invasion are described as a key difference, but there only 1 extra EVT cell in the normal pregnancy compared to the PE.

Author Response

Thank you for the thoughtful and constructive feedback you provided regarding our manuscript, “HMGA is a key molecule for the physiology of normal pregnancy and the pathogenesis of preeclampsia”. We agree with you that suggested to change and modulate my manuscript, and we have amended this by changing the contents in order to better conform

with the formatting and content rules of Biomolecules.

Reply on the 2nd reviewer,

#1. “On line 50-51, the authors refer to trophoblasts as physiological cancer cells. This may be misleading to some readers and confused for choriocarcinoma cells. I would suggest re-wording this.

              Ans1. Thank you for your advice. We removed the word.

#2. “Throughout the text, the authors list roles for HMGA in various cellular processes, but only superficially.”

              Ans2. Thank you for your advice. We added some sentences for the explanation in detail.

#3. “On line 239, the authors describe their own work, referring to their PE mouse model. Please elaborate on the details of this mouse model and the data that were generated from it.”

              Ans3. Thank you very much for your interest. We extracted some data from the manuscript and edited it.

#4. “but the images on the right half of the figure do not portray what is described on the left half.”

              Ans4. Thank you very much for your advice. We re-described the figure along your advice.

Round 2

Reviewer 1 Report

#1. “In their last sentence, the authors state that ‘HMGA1 is a potential driver of PE…’, this could be a title much more acceptable.”

              Ans1. Thank you for your advice. We change the title from original sentence to revised sentence “HMGA1 is a potential driver of preeclampsia pathogenesis by interference with extravillous trophoblasts invasion.”.

I agree much more with this new title

#2. “It seems that the direct involvement of HMGA in this transition is not well established”

              Ans2. Thank you for your suggestion. It is true and we have included new sentences concerning to the transition in the last paragraph.

The explanations lines 268-279 give a more balanced vision where proliferation is duly mentioned

#3. “Are there placental-specific co-factors interacting with HMGA proteins?”

              Ans3. Thank you for your suggestion. It is known that the intracellular signaling pathway of Wnt plays an important role in the function of HMGA1. We have added that explanation into the manuscript.

Sorry, can you tell me where you mention this in the text? I did not find mention of Wnt in the text, neither with specific partners or interactants.

#4. “a recent study, pointed to HMGN1 as being targeted by the preeclampsia-associated transcription factor STOX1 (Ducat et al, iScience, 2020).”

Ans4. We cited the reference and added new sentences about HMGN1.

Sorry, I did not find reference to HMGN1, nor the additional reference.

#5. “The authors should also cite (Li et al, Int J Obstet and Gynecol 2018 PMID: 29569400, that shows that HMGB1 is increased in preeclampsia sera”

              Ans5. We also cited the reference.

Sorry, I did not find reference to HMGB1in relation with the additional reference, neither did I find the aforementioned reference.

Author Response

I am very sorry. I was making a paragraph for each block in response to your advice and overlooked the final result.

Thank you very much.

#3. “Are there placental-specific co-factors interacting with HMGA proteins?”

              Ans3. Thank you for your suggestion. It is known that the intracellular signaling pathway of Wnt plays an important role in the function of HMGA1. We have added that explanation into the manuscript.

Sorry, can you tell me where you mention this in the text? I did not find mention of Wnt in the text, neither with specific partners or interactants.

Sorry. That part was left out while revising the paper. The part has been added on the line 225-233.

#4. “a recent study, pointed to HMGN1 as being targeted by the preeclampsia-associated transcription factor STOX1 (Ducat et al, iScience, 2020).”

Ans4. We cited the reference and added new sentences about HMGN1.

Sorry, I did not find reference to HMGN1, nor the additional reference.

Sorry. That part was left out while revising the paper. The part has been added on the line 279-290.

#5. “The authors should also cite (Li et al, Int J Obstet and Gynecol 2018 PMID: 29569400, that shows that HMGB1 is increased in preeclampsia sera”

              Ans5. We also cited the reference.

Sorry, I did not find reference to HMGB1in relation with the additional reference, neither did I find the aforementioned reference.

Sorry. That part was left out while revising the paper. The part has been added on the line 291-294.

Round 3

Reviewer 1 Report

Thank you for your answers to my issues.